# Communication Disparities between Nursing Home Team Members

**DOI:** 10.3390/ijerph19105975

**Published:** 2022-05-14

**Authors:** Timothy W. Farrell, Jorie M. Butler, Gail L. Towsley, Jacqueline S. Telonidis, Katherine P. Supiano, Caroline E. Stephens, Nancy M. Nelson, Alisyn L. May, Linda S. Edelman

**Affiliations:** 1Division of Geriatrics, Spencer Fox Eccles School of Medicine, University of Utah, 30 N 1900 E, AB 193 SOM, Salt Lake City, UT 84132, USA; jorie.butler@psych.utah.edu; 2Geriatric Research, Education, and Clinical Center (GRECC), George E. Wahlen Veteran Affairs Medical Center, 500 Foothill Drive, Salt Lake City, UT 84148, USA; 3College of Nursing, University of Utah, 10 S 2000 E, Salt Lake City, UT 84112, USA; gail.towsley@nurs.utah.edu (G.L.T.); jacqueline.telonidis@hsc.utah.edu (J.S.T.); katherine.supiano@nurs.utah.edu (K.P.S.); caroline.stephens@nurs.utah.edu (C.E.S.); nancy.nelson@nurs.utah.edu (N.M.N.); linda.edelman@nurs.utah.edu (L.S.E.); 4College of Pharmacy, University of Utah, 30 S 2000 E, Salt Lake City, UT 84112, USA; alisyn.may@pharm.utah.edu

**Keywords:** long-term care, nursing home, nurse staffing, interprofessional communication, care coordination, teamwork

## Abstract

Optimal care in nursing home (NH) settings requires effective team communication. Certified nursing assistants (CNAs) interact with nursing home residents frequently, but the extent to which CNAs feel their input is valued by other team members is not known. We conducted a cross-sectional study in which we administered a communication survey within 20 Utah nursing home facilities to 650 team members, including 124 nurses and 264 CNAs. Respondents used a 4-point scale to indicate the extent to which their input is valued by other team members when reporting their concerns about nursing home residents. We used a one-way ANOVA with a Bonferroni correction. When compared to nurses, CNAs felt less valued (CNA mean = 2.14, nurse mean = 3.24; *p* < 0.001) when reporting to physicians, and less valued (CNA mean = 1.66, nurse mean = 2.71; *p* < 0.001) when reporting to pharmacists. CNAs did not feel less valued than nurses (CNA mean = 3.43, nurse mean = 3.37; *p* = 0.25) when reporting to other nurses. Our findings demonstrate that CNAs feel their input is not valued outside of nursing, which could impact resident care. Additional research is needed to understand the reasons for this perception and to design educational interventions to improve the culture of communication in nursing home settings.

## 1. Introduction

Person-centered care is defined as “providing care that is respectful of and responsive to individual patient preferences, needs, and values, and ensuring that patient values guide all clinical decisions and recognizes that preferences extend beyond medical and clinical concerns [1,2]”. Optimal person-centered care in nursing homes (NH) requires strong interprofessional teamwork [3,4,5,6]. Good communication is a hallmark of effective teams [7,8]. Quality improvement programs such as INTERACT^®^ (Interventions to Improve Acute Care Transfers) aim to improve communication in long-term care settings [9]. At its best, strong team communication in NHs results in care delivery aligned with age-friendly care, including basing the care plan on what matters most to residents and their caregivers and contributes to improved patient safety [1,10,11]. This approach also fosters a culture in which all team members are encouraged to report changes in condition [12]. Timely reporting of changes in condition can prevent unnecessary care transitions [9,13].

Communication in NH settings is often far from optimal, with many reports in the literature discussing poor or ineffective communication between team members and with residents and their families [14,15,16,17]. Poor communication contributes to poor outcomes [13,18,19,20,21]. For example, inadequate communication between team members and residents and families around goals of care can contribute to poor care transitions and fragmentation of care that result in decreased care quality and satisfaction [13,19,22,23]. Further, poor communication has also been associated with increased job dissatisfaction and staff turnover [24,25,26,27].

Person-centered care is a key component of the Medicare and Medicaid survey process [28,29] and is paramount to staff practices. A key component of person-centered care is strong team communication [2,30,31]. Numerous barriers to optimal person-centered communication in NHs include lack of respect and empathy between staff members [32], lack of clarity around role expectations [33,34], as well as limited staff training in person-centered care and communication skills [35,36]. Another barrier may be the culture of NH facilities [19,27,37,38]. It has been shown that NH culture may influence staff members’ self-efficacy in communicating with colleagues [39,40]. Staff self-efficacy may be related to how valued they feel by team members, especially those from other job groups or professions.

It is important to understand how the interprofessional NH team communicates, particularly nurses and CNAs who work most closely with residents [34,41] and are thus most likely to be present when a change in condition occurs. Nurse and CNA communication with each other and other team members is critical at these times. However, communication between nurses, who supervise CNAs and communicate with other clinicians, and CNAs, who generally are among the lowest paid and least experienced NH staff, may be ineffective if nurses are not open to communication or if CNAs feel their input is not valued. Similarly, nurses may hesitate to communicate with other clinicians if they do not feel that their professional input is taken seriously. This paper explores how RNs and CNAs value communication with each other and other members of the interprofessional NH team.

## 2. Materials and Methods

After gaining permission from administrators for this cross-sectional study, staff members from 20 NH facilities in Utah were recruited between December 2017 and June 2018 to complete a pen and paper survey during quality improvement or all-staff meetings. Participants were provided with a survey cover letter. Surveys were anonymous and no personal identifiers were captured. The University of Utah IRB deemed this study exempt, waiving the need for signed informed consent.

Each participant completed a long-term care communication survey that included demographics and questions organized according to four domains: (1) values/ethics for interprofessional practice; (2) sensory deficits (hearing loss and vision loss); (3) health literacy, and (4) effective communication. We previously described the development of this survey and established its content validity [42].

For this paper, we focused on the responses of CNAs and nurses to the first domain (values/ethics for interprofessional practice) survey items that addressed the extent to which they felt valued by other job groups in their facility. The item responses were on a 4-point scale (1 = not valued, 2 = somewhat valued, 3 = mostly valued, 4 = highly valued) (see Appendix A for the survey question).

### Statistical Analysis

Statistical analyses were performed in SPSS v26. We performed one-way Analyses of Variance (ANOVA) with a Bonferroni correction to account for the multiple comparisons among job groups. The Bonferroni correction confirmed that p values less than 0.05 were true and not by chance. We then compared CNAs to nurses regarding the extent to which feedback about residents was perceived as valued by other team members. We also conducted a t-test comparing CNAs’ and nurses’ perceptions of value when reporting to other team members.

## 3. Results

Among the 650 survey respondents, there were 264 (40.6%) CNAs, including two medical technicians, and 124 (19.1%) nurses, including 96 clinical nurses, one RN supervisor, 26 licensed nursing/Minimum Data Set coordinators, and one unit manager.

The demographics of the survey respondents are listed in Table 1. Respondents were overwhelmingly female and white. Nearly 1/5th of CNAs (19.7%) reported Hispanic ethnicity, compared to nearly 1/10th of nurses (9.7%) that reported Hispanic ethnicity. One-third (33.3%) of CNAs reported no college education, while 79.8% of nurses were college graduates.

Nurses’ and CNAs’ sense of value in reporting to three professional job groups (physicians, pharmacists, and nurses) are shown in Table 2. Nurses reported feeling their reporting concerns were somewhat valued by pharmacists (X = 2.71) and most valued by physicians (X = 3.24) and other nurses (X = 3.37). Conversely, CNAs reported their concerns were not valued by pharmacists (X = 1.66), somewhat valued by physicians (X = 1.66), and most valued by nurses (X = 3.42). Between-group differences in CNA reports versus nurses’ reports were detected in reports to physicians (F(8) = 7.13; *p* < 0.01) and to pharmacists (F(8) = 1.29, *p* = 0.025), but not to nurses (F(8) = 1.29, *p* = 0.25). Thus, CNAs, on average, reported feeling less valued by other professions than nursing when reporting their concerns about residents (see Table 2).

## 4. Discussion

We deployed a communication survey among NH care team members to compare the extent to which CNAs and nurses feel their input on resident conditions is valued by other team members (physicians, pharmacists, and nurses). Our findings show that both nurses and CNAs feel that their input is less valued by pharmacists and physicians than nurses.

It was not surprising that CNAs felt less valued than nurses when reporting their concerns about residents to physicians and pharmacists. Physicians and pharmacists may undervalue CNAs input in NHs. This could be in part due to physicians and pharmacists having little or no exposure to the CNA role in general or due to the predominant role they play in NH settings during their undergraduate and graduate education. Even when working with NH residents, they may spend little time in the NH setting and infrequently communicate with CNAs. Therefore, they may not observe the role of CNAs or recognize that CNAs spend more time than any other job title interacting directly with residents. Practicing physicians and pharmacists may, therefore, be less likely to recognize CNAs as an important member of the NH care team.

Even though CNAs spend the most time with residents, they lack autonomy and power [26,34,43,44,45]. The rapid turnover of CNAs, with the majority staying in one position for less than one year [14,35], may contribute to the comfort CNAs have in communicating with physicians and pharmacists. Programs like INTERACT^®^ [9] that utilize communication tools such as Stop and Watch encourage all staff to report changes in resident status. However, these changes are most often reported to nurses rather than other providers. Nonetheless, utilizing tools that empower CNAs to communicate effectively and that encourage teams to listen to all voices may be a way to build the sense of value that CNAs perceive others place on their input.

Interestingly, our results show that CNAs perceive that nurses value their input about NH residents as much as nurses perceive other nurses do. Intuitively, there are several reasons for this. Nurses are familiar with CNA roles in the NH setting and rely on them to be their eyes and ears regarding resident status and wellbeing [35]. In addition, some of the survey respondents who are nurses may have been CNAs before obtaining their nursing degrees. It is encouraging that CNAs in this study perceived their input was valued by the nurses they work with, suggesting that despite the general lack of leadership training for nurses in delegating tasks to CNAs, they are communicating as a team to provide care for residents. Other disciplines would do well to look to the example of nursing in valuing input for CNAs. This may require both discipline-specific and cross-disciplinary training.

Person-centered care is an expectation of today’s NHs [2,29,31,46]. In order for this to occur, it is imperative that there is effective communication between all NH team members, including CNAs [32,47,48]. This can be difficult for CNAs, whose workflow necessitates that word-of-mouth and informal face-to-face conversation is the most expedient way to share information [47]. Nurses are more able to engage in these types of communication with CNAs, which may be why, in our study, CNAs felt their input was more valued by nurses than physicians and pharmacists. Because communication between CNAs and nurses tends to be informal and is likely to be more frequent, it is important that nurses then take the results of the communication exchange to other members of the team. Training nurses to communicate expediently and effectively with others using tools such as SBAR (Situation, Background, Assessment, Recommendation) can improve resident safety [13].

Although our data were obtained prior to the COVID pandemic, we feel that our findings have even greater relevance since the pandemic. Nurses and CNAs are the backbone of the NH response to COVID. Even before the pandemic, NHs were challenged to hire and retain enough nurses and CNAs due to low wages and challenging working conditions [26,45,49,50,51]. These challenges have been amplified during the pandemic. An emphasis on changing NH culture to value the contributions of nurses and CNAs by improving communication could improve NH response to the needs of residents during the pandemic, such as infection control practices and nurse and CNA retention, thereby improving resident outcomes.

Our study has several limitations. First, NH residents and staff did not contribute to survey development. As a result, they may have perceived that the survey questions were not directly applicable to their roles and responsibilities in their facility. In addition, all 20 NH facilities are located in Utah. As such, the generalizability of our findings may be limited both with respect to other states in the U.S. and also internationally. This may be especially so because Utah has less racial and ethnic diversity than many other states. In addition, we did not measure how RNs and CNAs felt their input was valued by residents and families. A hallmark of person-centered care is communication that goes beyond discussing tasks with residents to focus on what matters to individual residents leading to shared decision making [2,15]. Understanding how CNAs perceive their input is valued by residents and families could be used to develop educational materials for CNAs, residents, and families about fostering better communication.

In our future work, we plan to explore several new questions raised by these findings. First, it is unclear why CNAs feel more undervalued by physicians and pharmacists than nurses do and if this perception impacts CNA retention and resident outcomes. Second, that both CNAs and nurses perceived their input was the least valued by pharmacists over physicians and other nurses are worthy of further exploration. Third, the quality of NH staff communication may be an attractive metric to incorporate in nursing home quality measures. Finally, our findings have implications for interprofessional education [7], which is ideally positioned to teach health sciences students about the roles and responsibilities of team members, as well as providers who practice in NH settings.

## 5. Conclusions

CNAs felt less valued than nurses when reporting their concerns about residents to physicians and pharmacists but not to nurses. This discrepancy suggests opportunities for interprofessional team training in NH facilities. Additional research is needed to understand the reasons for variations in nurses’ and CNAs’ perceptions that their input about NH residents is undervalued by some team members, particularly pharmacists, and to design educational interventions during training and in the workplace to improve team communication in NH settings. These trainings will take on added urgency given the stress that the COVID pandemic is currently placing on the NH nursing and CNA workforces.

## Figures and Tables

**Table 1 ijerph-19-05975-t001:** Demographics of Survey Respondents.

Characteristic	CNAs	Nurses
	*n* (%)	*n* (%)
Gender		
Male	46 (17.4)	14 (11.3)
Female	210 (79.5)	106 (85.5)
Transgender	1 (0.4)	0 (0.0)
Preferred not to respond	2 (0.8)	1 (0.8)
Age		
Under 25 years old	127 (50.8%)	13 (11.3%)
25 to 34 years old	52 (20.8%)	31 (27.0%)
35 to 44 years old	37 (14.8%)	30 (26.1%)
45 to 54 years old	21 (8.4%)	19 (16.5%)
55 to 64 years old	10 (4.0%)	16 (13.9%)
65 years old and above	3 (1.2%)	6 (5.2%)
Race		
American Indian or Alaska Native	8 (3.0)	3 (2.4)
Asian	12 (4.5)	4 (3.2)
Black or African American	5 (1.9)	0 (0.0)
Native Hawaiian/Other Pacific Islander	11 (4.2)	0 (0.0)
White	181 (68.6)	105 (84.7)
Other	9 (3.4)	1 (0.8)
Ethnicity		
Hispanic, Latino, or of Spanish Origin	52 (19.7)	12 (9.7)
Non-Hispanic or Non-Latino	199 (75.4)	107 (86.3)
Education		
8th Grade or Less	1 (0.4)	0 (0.0)
Some High School	13 (4.9)	0 (0.0)
High School Graduate	74 (28.0)	0 (0.0)
Some College	130 (49.2)	14 (11.3)
College Graduate	38 (14.4)	99 (79.8)
Postgrad/Professional	3 (1.1)	6 (4.8)
Total Time Worked at Facility		
Fewer than 6 months	48 (19.7%)	15 (12.8%)
6 months to less than 1 year	55 (22.5%)	10 (8.5%)
1 year to fewer than 2 years	48 (19.7%)	14 (12.0%)
2–5 years	61 (25.0%)	48 (41.0%)
6–10 years	22 (9.0%)	22 (18.8%)
11–20 years	7 (2.9%)	4 (3.4%)
More than 20 years	3 (1.2%)	4 (3.4%)

Number of totals may not equal 100% due to participant non-response.

**Table 2 ijerph-19-05975-t002:** Nurses and CNAs’ Perceived Sense of Value when Reporting to Physicians, Pharmacists, and Nurses.

Job Group	Extent to Which I Felt Valued Reporting Concerns about Residents to Physicians	Extent to Which I Felt Valued Reporting Concerns about Residents to Pharmacists	Extent to Which I Felt Valued Reporting Concerns about Residents to Nurses
	Mean (SD)	Mean (SD)	Mean (SD)
Nurses	3.24 (0.90) *	2.71 (1.34) *	3.37 (0.83)
CNAs	2.14 (1.74) *	1.66 (1.70) *	3.42 (0.79)

* *p* < 0.05.

## Data Availability

The data presented in this study are available upon request from the corresponding author.

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
