# Peer review of "Communication Disparities between Nursing Home Team Members"

_ijerph, 2022, doi:10.3390/ijerph19105975_

Round 1

Reviewer 1 Report

I found it very interesting to read this manuscript.

The Introduction is rather good, but it needed more about the key concepts such as Covid-19. It was strange to see that only in the discussion chapter. Person-centered care can also be described in the introduction in more detail. Why is the keyword age-friendly care?

Materials and Methods

It looks like chapter 3 results is missing. There is one table in chapter 2 with results.

Don’t you have more results to publish than that?

The data was gathered 2017-2018. If you want to relate it to the pandemic why is the pandemic not mentioned at the end of the introduction as part of the research question?

 I would have liked to have more description of the material and method. It would be interesting to see more about the participants e.g. how many they were, how many in each facilities, age groups, years in practice in the nursing home etc. Better description about the data analysis.

You do not mention any ethical concerns or any need to take notice of that matter.

Discussion

There is something about INTERACT? Why not mention that before for example in the introduction?

There COVID-19 is also mentioned and is part of the keywords. Why are they part of your keywords? The data was obtained prior to the pandemic.

Author Response

Reviewer 1 Comment

Authors’ Response

Relevant Page, Section, and Paragraph Number

The Introduction is rather good, but it needed more about the key concepts such as Covid-19. It was strange to see that only in the discussion chapter.

Person-centered care can also be described in the introduction in more detail.

Why is the keyword age-friendly care?

Thank you for the question about where to comment on COVID-19. We respectfully disagree that COVID-19 should be mentioned in the introduction because this study was designed and implemented before the pandemic.  We therefore believe it would be odd to mention COVID in the introduction.  We feel that it is most appropriate to discuss COVID only in the Discussion section where we state “Although our data were obtained prior to the COVID pandemic, we feel that our findings have even greater relevance since the pandemic.”

Thank you for this comment.  We have added the National Academy of Medicine’s definition of person-centered care to the first paragraph of the introduction. 

We have removed age-friendly care from the key words since it is not a central concept in the manuscript.

Page 6, Section 4, Paragraph 6 in attached manuscript with Track Changes.

Page 1, Section 1, Paragraph 1 in attached manuscript with Track Changes.

Page 1, Abstract, in attached manuscript with Track Changes.

It looks like chapter 3 results is missing. There is one table in chapter 2 with results.

Don’t you have more results to publish than that?

The Statistical Analysis paragraph at the end of the Materials and Methods section as well as the Results section (section 3) were submitted but not included in the version that went to reviewers.  We have added back these sections here.

Pages 2-4, Sections 2--3, in attached manuscript with Track Changes.

The data was gathered 2017-2018. If you want to relate it to the pandemic why is the pandemic not mentioned at the end of the introduction as part of the research question?

Thank you for this question.  We do not wish to relate the pandemic to the data because the data were collected prior to the pandemic.  However, we do feel it is appropriate to state that our findings have increased relevance after the pandemic because the importance of improving communication in long-term settings was important prior to the pandemic but has taken on greater significance post-pandemic given the substantial toll of the pandemic on the long-term care workforce.

Page 6, Section 4, Paragraph 6 in attached manuscript with Track Changes.

 I would have liked to have more description of the material and method. It would be interesting to see more about the participants e.g. how many they were, how many in each facilities, age groups, years in practice in the nursing home etc. Better description about the data analysis.

Table 1 was submitted but not included in the version that went to reviewers.  We have reinserted this table, which presents demographic data including age, years in practice, gender, race, ethnicity, education.  Because some facilities had few respondents, we did not report results by facility in order to retain participants’ anonymity.

Page 3, Section 3 in attached manuscript with Track Changes.

You do not mention any ethical concerns or any need to take notice of that matter.

Thank you for this comment.  We do not believe our study has any ethical concerns.  As stated in the Materials and Methods section, “Surveys were anonymous and no personal identifiers were captured. The University of Utah IRB deemed this study exempt, waiving the need for signed informed consent.” 

Page 2, Section 2, Paragraph 1 in attached manuscript with Track Changes.

There is something about INTERACT? Why not mention that before for example in the introduction?

Thank you for this comment.  We have added a sentence about INTERACT to the first paragraph of the introduction section.

Page 1, Section 1, Paragraph 1 in attached manuscript with Track Changes.

There COVID-19 is also mentioned and is part of the keywords. Why are they part of your keywords? The data was obtained prior to the pandemic.

Thank you for this comment.  We agree and have removed COVID-19 from the keywords.

Page 1, Abstract in attached manuscript with Track Changes.

Reviewer 2 Report

I want to congratulate the authors for the theme of the manuscript. In my opinion, I found the study very interesting and I think the topic is very necessary.

Abstract:

Adequate background, but there is an excessive use of abbreviations. The type of study is not clear.

The title: too long and The descriptors (key words) of the study are not clearly identified.

 Introduction:

References in the text and adequate justification of data. The international scope and a vision from the most general to the most specific would be lacking.

Materials and methods:

- The design of the study and the type of sampling criteria for inclusion, exclusion, losses, ... are not well defined.

Study design and sample

It would be necessary to specify the type of study and how the survey was carried out. In general, there is a lack of information on data collection, procedures and measurement of the different types of variables. The variables are not clear enough.

 The programs used in data analysis, the size of the sample calculation, and the validity and reliability of the instrument used are not correctly defined.

Results

The presentation and design of the results is not correct, the absence of figures and the inappropriate use of tables.

Discussion

-The order in the presentation of the discussion is adequate.

Non-current references.

Author Response

Reviewer 2 Comment

Authors’ Response

Relevant Page, Section, and Paragraph Number

Adequate background, but there is an excessive use of abbreviations.

The type of study is not clear.

Thank you for this comment.  We reduced the number of abbreviations in the abstract by writing out “nursing home.”  We felt it made sense to keep the CNA abbreviation because this acronym is in common use to describe certified nursing assistants.

We added text to the abstract to indicate that this is a cross-sectional study.

Page 1, Abstract in attached manuscript with Track Changes

Page 1, Abstract in attached manuscript with Track Changes

The title: too long and The descriptors (key words) of the study are not clearly identified.

Thank you for this comment.  We have suggested an edit to reduce the length of the title.  We changed the title from “Gaining Insights into Nursing Home Culture:  Communication Disparities between Certified Nursing Assistants, Physician Assistants, Physicians, Pharmacists, and Nurses” to “Communication Disparities between Nursing Home Team Members”

We also removed 2 key words (age-friendly care and COVID-19) that are not central to the manuscript.  We believe that the remaining key words all relate to the core ideas conveyed in the manuscript.

Page 1, Title in attached manuscript with Track Changes

Page 1, Abstract in attached manuscript with Track Changes

References in the text and adequate justification of data.

The international scope and a vision from the most general to the most specific would be lacking.

Thank you, we agree that we have provided ample references in the text and provided rationale for the data.

We acknowledge in the Discussion section that the generalizability of our findings may be limited because the study was conducted in the United States (in the state of Utah).  We have edited the Discussion accordingly.

We are not sure what Reviewer 2 means by stating that “a vision from the most general to the most specific would be lacking.”  We speculate that Reviewer 2 may be asking if we have data beyond the state of Utah that we could then use to make more local inferences, but since our data source was in Utah only, we cannot make these inferences.

Page 7-9, References in attached manuscript with Track Changes

Page 6, Section 4, Paragraph 7 in attached manuscript with Track Changes

Page 6, Section 4, Paragraph 7 in attached manuscript with Track Changes

The design of the study and the type of sampling criteria for inclusion, exclusion, losses, ... are not well defined.

We added text to the Materials and Methods section to indicate that this is a cross-sectional study.

The statistical analysis approach and results sections were submitted but not included in the version that went to reviewers, hence the confusion about missing materials.  This has been reinserted in the reviewer document and should provide needed information to address reviewers’ concerns.

Page 2, Section 2, Paragraph 1 in attached manuscript with Track Changes

It would be necessary to specify the type of study and how the survey was carried out. In general, there is a lack of information on data collection, procedures and measurement of the different types of variables. The variables are not clear enough.

As noted above, the statistical analysis approach and results sections were submitted but not included in the version that went to reviewers, hence the confusion about missing materials.  This has been reinserted in the reviewer document and should provide needed information to address reviewers’ concerns.

Pages 2-4, Sections 2--3 in attached manuscript with Track Changes

 The programs used in data analysis, the size of the sample calculation, and the validity and reliability of the instrument used are not correctly defined.

As noted above, the statistical analysis approach and results sections were submitted but not included in the version that went to reviewers, hence the confusion about missing materials.  This has been reinserted in the reviewer document and should provide needed information to address reviewers’ concerns.

Pages 2-4, Sections 2--3 in attached manuscript with Track Changes

The presentation and design of the results is not correct, the absence of figures and the inappropriate use of tables.

The statistical analysis approach and results sections were submitted but not included in the version that went to reviewers, hence the confusion about missing materials.  This has been reinserted in the reviewer document and should provide needed information to address reviewers’ concerns.

Pages 2-4, Sections 2--3 in attached manuscript with Track Changes

The order in the presentation of the discussion is adequate.

Thank you for this feedback.

Pages 5-6, Section 4 in attached manuscript with Track Changes

Non-current references.

We believe we have an adequate number of current references.  8/48 references were from the last 3 years (2020 – 2022), and 17/48 references were from 2015 – 2019.  Overall, 25/48 of the references we cited were no more than 7 years old.

Page 7-9, References in attached manuscript with Track Changes

Round 2

Reviewer 1 Report

I found the manuscript much better now. It is interesting and I wish you all the very best. I have no further comments.

Author Response

Thank you for making the time to review the manuscript, Communication Disparities between Nursing Home Team Members. We appreciate your feedback!

Reviewer 2 Report

The bibliography should be updated, very old citations

Author Response

We reviewed and replaced all citations before year 2015 (please see the attachment), except for these key citations which are foundational:

  • Institute of Medicine Committee on Quality of Health Care in America. In Crossing the Quality Chasm: A New Health System for the 21st Century; National Academies Press (US): Washington (DC), 2001.
  • Ouslander, J.G.; Bonner, A.; Herndon, L.; Shutes, J. The Interventions to Reduce Acute Care Transfers (INTERACT) quality improvement program: an overview for medical directors and primary care clinicians in long term care. J Am Med Dir Assoc 2014, 15, 162-170, doi:10.1016/j.jamda.2013.12.005.
  • King, B.J.; Gilmore-Bykovskyi, A.L.; Roiland, R.A.; Polnaszek, B.E.; Bowers, B.J.; Kind, A.J. The consequences of poor communication during transitions from hospital to skilled nursing facility: a qualitative study. J Am Geriatr Soc 2013, 61, 1095-1102, doi:10.1111/jgs.12328.
  • Miller, S.C.; Looze, J.; Shield, R.; Clark, M.A.; Lepore, M.; Tyler, D.; Sterns, S.; Mor, V. Culture change practice in U.S. Nursing homes: prevalence and variation by state medicaid reimbursement policies. Gerontologist 2014, 54, 434-445, doi:10.1093/geront/gnt020.
  • Koren, M.J. Person-centered care for nursing home residents: the culture-change movement. Health Aff (Millwood) 2010, 29, 312-317, doi:10.1377/hlthaff.2009.0966.
  • Siegel, E.O.; Young, H.M. Communication between nurses and unlicensed assistive personnel in nursing homes: explicit expectations. J Gerontol Nurs 2010, 36, 32-37, doi:10.3928/00989134-20100702-02.
  • Tjia, J.; Mazor, K.M.; Field, T.; Meterko, V.; Spenard, A.; Gurwitz, J.H. Nurse-physician communication in the long-term care setting: perceived barriers and impact on patient safety. J Patient Saf 2009, 5, 145-152, doi:10.1097/PTS.0b013e3181b53f9b.
  • Cadogan, M.P.; Franzi, C.; Osterweil, D.; Hill, T. Barriers to effective communication in skilled nursing facilities: differences in perception between nurses and physicians. J Am Geriatr Soc 1999, 47, 71-75, doi:10.1111/j.1532-5415.1999.tb01903.x.
  • Kane, R.A. Ethics and the frontline care worker: mapping the subject. Generations 1994, 18, 71-74.

Overall, we now have 42 citations added to manuscript all dated 2015 or more recent.
